# Online Gradient Boosting

**Alina Beygelzimer**
Yahoo Labs
New York, NY 10036
beygel@yahoo-inc.com

**Elad Hazan**
Princeton University
Princeton, NJ 08540
ehazan@cs.princeton.edu

**Satyen Kale**
Yahoo Labs
New York, NY 10036
satyen@yahoo-inc.com

**Haipeng Luo**
Princeton University
Princeton, NJ 08540
haipengl@cs.princeton.edu

## Abstract

We extend the theory of boosting for regression problems to the online learning setting. Generalizing from the batch setting for boosting, the notion of a weak learning algorithm is modeled as an online learning algorithm with linear loss functions that competes with a base class of regression functions, while a strong learning algorithm is an online learning algorithm with smooth convex loss functions that competes with a larger class of regression functions. Our main result is an online gradient boosting algorithm that converts a weak online learning algorithm into a strong one where the larger class of functions is the linear span of the base class. We also give a simpler boosting algorithm that converts a weak online learning algorithm into a strong one where the larger class of functions is the convex hull of the base class, and prove its optimality.

## 1 Introduction

Boosting algorithms [21] are ensemble methods that convert a learning algorithm for a base class of models with weak predictive power, such as decision trees, into a learning algorithm for a class of models with stronger predictive power, such as a weighted majority vote over base models in the case of classification, or a linear combination of base models in the case of regression.

Boosting methods such as AdaBoost [9] and Gradient Boosting [10] have found tremendous practical application, especially using decision trees as the base class of models. These algorithms were developed in the batch setting, where training is done over a fixed batch of sample data. However, with the recent explosion of huge data sets which do not fit in main memory, training in the batch setting is infeasible, and online learning techniques which train a model in one pass over the data have proven extremely useful.

A natural goal therefore is to extend boosting algorithms to the online learning setting. Indeed, there has already been some work on online boosting for classification problems [20, 11, 17, 12, 4, 5, 2]. Of these, the work by Chen et al. [4] provided the first theoretical study of online boosting for classification, which was later generalized by Beygelzimer et al. [2] to obtain optimal and adaptive online boosting algorithms.

However, extending boosting algorithms for regression to the online setting has been elusive and escaped theoretical guarantees thus far. In this paper, we rigorously formalize the setting of online boosting for regression and then extend the very commonly used gradient

boosting methods [10, 19] to the online setting, providing theoretical guarantees on their performance.

The main result of this paper is an online boosting algorithm that competes with any linear combination the base functions, given an online linear learning algorithm over the base class. This algorithm is the online analogue of the batch boosting algorithm of Zhang and Yu [24], and in fact our algorithmic technique, when specialized to the batch boosting setting, provides exponentially better convergence guarantees.

We also give an online boosting algorithm that competes with the best convex combination of base functions. This is a simpler algorithm which is analyzed along the lines of the Frank-Wolfe algorithm [8]. While the algorithm has weaker theoretical guarantees, it can still be useful in practice. We also prove that this algorithm obtains the optimal regret bound (up to constant factors) for this setting.

Finally, we conduct some proof-of-concept experiments which show that our online boosting algorithms do obtain performance improvements over different classes of base learners.

## 1.1 Related Work

While the theory of boosting for classification in the batch setting is well-developed (see [21]), the theory of boosting for regression is comparatively sparse.The foundational theory of boosting for regression can be found in the statistics literature [14, 13], where boosting is understood as a greedy stagewise algorithm for fitting of additive models. The goal is to achieve the performance of linear combinations of base models, and to prove convergence to the performance of the best such linear combination.

While the earliest works on boosting for regression such as [10] do not have such convergence proofs, later works such as [19, 6] do have convergence proofs but without a bound on the speed of convergence. Bounds on the speed of convergence have been obtained by Duffy and Helmbold [7] relying on a somewhat strong assumption on the performance of the base learning algorithm. A different approach to boosting for regression was taken by Freund and Schapire [9], who give an algorithm that reduces the regression problem to classification and then applies AdaBoost; the corresponding proof of convergence relies on an assumption on the induced classification problem which may be hard to satisfy in practice. The strongest result is that of Zhang and Yu [24], who prove convergence to the performance of the best linear combination of base functions, along with a bound on the rate of convergence, making essentially no assumptions on the performance of the base learning algorithm. Telgarsky [22] proves similar results for logistic (or similar) loss using a slightly simpler boosting algorithm.

The results in this paper are a generalization of the results of Zhang and Yu [24] to the online setting. However, we emphasize that this generalization is nontrivial and requires different algorithmic ideas and proof techniques. Indeed, we were not able to directly generalize the analysis in [24] by simply adapting the techniques used in recent online boosting work [4, 2], but we made use of the classical Frank-Wolfe algorithm [8]. On the other hand, while an important part of the convergence analysis for the batch setting is to show statistical consistency of the algorithms [24, 1, 22], in the online setting we only need to study the empirical convergence (that is, the regret), which makes our analysis much more concise.

## 2 Setup

Examples are chosen from a feature space $\mathcal{X}$, and the prediction space is $\mathbb{R}^d$. Let $\|\cdot\|$ denote some norm in $\mathbb{R}^d$. In the setting for online regression, in each round $t$ for $t = 1, 2, \ldots, T$, an adversary selects an example $\mathbf{x}_t \in \mathcal{X}$ and a loss function $\ell_t : \mathbb{R}^d \to \mathbb{R}$, and presents $\mathbf{x}_t$ to the online learner. The online learner outputs a prediction $\mathbf{y}_t \in \mathbb{R}^d$, obtains the loss function $\ell_t$, and incurs loss $\ell_t(\mathbf{y}_t)$.

Let $\mathcal{F}$ denote a reference class of regression functions $f : \mathcal{X} \to \mathbb{R}^d$, and let $\mathcal{C}$ denote a class of loss functions $\ell : \mathbb{R}^d \to \mathbb{R}$. Also, let $R : \mathbb{N} \to \mathbb{R}_+$ be a non-decreasing function. We say that the function class $\mathcal{F}$ is *online learnable* for losses in $\mathcal{C}$ with regret $R$ if there is an online learning algorithm $\mathcal{A}$, that for every $T \in \mathbb{N}$ and every sequence $(\mathbf{x}_t, \ell_t) \in \mathcal{X} \times \mathcal{C}$ for

$t = 1, 2, \ldots, T$ chosen by the adversary, generates predictions[1] $\mathcal{A}(\mathbf{x}_t) \in \mathbb{R}^d$ such that

$$\sum_{t=1}^{T} \ell_t(\mathcal{A}(\mathbf{x}_t)) \ \leq \ \inf_{f \in \mathcal{F}} \sum_{t=1}^{T} \ell_t(f(\mathbf{x}_t)) + R(T). \tag{1}$$

If the online learning algorithm is randomized, we require the above bound to hold with high probability.

The above definition is simply the online generalization of standard empirical risk minimization (ERM) in the batch setting. A concrete example is 1-dimensional regression, i.e. the prediction space is $\mathbb{R}$. For a labeled data point $(\mathbf{x}, y^\star) \in \mathcal{X} \times \mathbb{R}$, the loss for the prediction $y \in \mathbb{R}$ is given by $\ell(y^\star, y)$ where $\ell(\cdot, \cdot)$ is a fixed loss function that is convex in the second argument (such as squared loss, logistic loss, etc). Given a batch of $T$ labeled data points $\{(\mathbf{x}_t, y_t^\star) \mid t = 1, 2, \ldots, T\}$ and a base class of regression functions $\mathcal{F}$ (say, the set of bounded norm linear regressors), an ERM algorithm finds the function $f \in \mathcal{F}$ that minimizes $\sum_{t=1}^{T} \ell(y_t^\star, f(\mathbf{x}_t))$.

In the online setting, the adversary reveals the data $(\mathbf{x}_t, y_t^\star)$ in an online fashion, only presenting the true label $y_t^\star$ after the online learner $\mathcal{A}$ has chosen a prediction $y_t$. Thus, setting $\ell_t(y_t) = \ell(y_t^\star, y_t)$, we observe that if $\mathcal{A}$ satisfies the regret bound (1), then it makes predictions with total loss almost as small as that of the empirical risk minimizer, up to the regret term. If $\mathcal{F}$ is the set of all bounded-norm linear regressors, for example, the algorithm $\mathcal{A}$ could be online gradient descent [25] or online Newton Step [16].

At a high level, in the batch setting, "boosting" is understood as a procedure that, given a batch of data and access to an ERM algorithm for a function class $\mathcal{F}$ (this is called a "weak" learner), obtains an approximate ERM algorithm for a richer function class $\mathcal{F}'$ (this is called a "strong" learner). Generally, $\mathcal{F}'$ is the set of finite linear combinations of functions in $\mathcal{F}$. The efficiency of boosting is measured by how many times, $N$, the base ERM algorithm needs to be called (i.e., the number of boosting steps) to obtain an ERM algorithm for the richer function within the desired approximation tolerance. Convergence rates [24] give bounds on how quickly the approximation error goes to 0 and $N \to \infty$.

We now extend this notion of boosting to the online setting in the natural manner. To capture the full generality of the techniques, we also specify a class of loss functions that the online learning algorithm can work with. Informally, an online boosting algorithm is a reduction that, given access to an online learning algorithm $\mathcal{A}$ for a function class $\mathcal{F}$ and loss function class $\mathcal{C}$ with regret $R$, and a bound $N$ on the total number of calls made in each iteration to copies of $\mathcal{A}$, obtains an online learning algorithm $\mathcal{A}'$ for a richer function class $\mathcal{F}'$, a richer loss function class $\mathcal{C}'$, and (possibly larger) regret $R'$. The bound $N$ on the total number of calls made to all the copies of $\mathcal{A}$ corresponds to the number of boosting stages in the batch setting, and in the online setting it may be viewed as a resource constraint on the algorithm. The efficacy of the reduction is measured by $R'$ which is a function of $R$, $N$, and certain parameters of the comparator class $\mathcal{F}'$ and loss function class $\mathcal{C}'$. We desire online boosting algorithms such that $\frac{1}{T} R'(T) \to 0$ quickly as $N \to \infty$ and $T \to \infty$. We make the notions of richness in the above informal description more precise now.

**Comparator function classes.** A given function class $\mathcal{F}$ is said to be $D$-bounded if for all $\mathbf{x} \in \mathcal{X}$ and all $f \in \mathcal{F}$, we have $\|f(\mathbf{x})\| \leq D$. Throughout this paper, we assume that $\mathcal{F}$ is symmetric:[2] i.e. if $f \in \mathcal{F}$, then $-f \in \mathcal{F}$, and it contains the constant zero function, which we denote, with some abuse of notation, by $\mathbf{0}$.

Given $\mathcal{F}$, we define two richer function classes $\mathcal{F}'$: the convex hull of $\mathcal{F}$, denoted $\mathrm{CH}(\mathcal{F})$, is the set of convex combinations of a finite number of functions in $\mathcal{F}$, and the span of $\mathcal{F}$, denoted $\mathrm{span}(\mathcal{F})$, is the set of linear combinations of finitely many functions in $\mathcal{F}$. For any $f \in \mathrm{span}(\mathcal{F})$, define $\|f\|_1 := \inf\left\{\max\{1, \sum_{g \in S} |w_g|\} : f = \sum_{g \in S} w_g g, \ S \subseteq \mathcal{F}, \ |S| < \infty, \ w_g \in \mathbb{R}\right\}$. Since functions in $\mathrm{span}(\mathcal{F})$ are not bounded, it is not possible to obtain a uniform regret bound for all functions in $\mathrm{span}(\mathcal{F})$: rather, the regret of an online learning algorithm $\mathcal{A}$ for $\mathrm{span}(\mathcal{F})$ is specified in terms of regret bounds for individual comparator functions $f \in \mathrm{span}(F)$, viz.

$$R_f(T) := \sum_{t=1}^{T} \ell_t(\mathcal{A}(\mathbf{x}_t)) - \sum_{t=1}^{T} \ell_t(f(\mathbf{x}_t)).$$

**Loss function classes.** The base loss function class we consider is $\mathcal{L}$, the set of all linear functions $\ell : \mathbb{R}^d \to \mathbb{R}$, with Lipschitz constant bounded by 1. A function class $\mathcal{F}$ that is online learnable with the loss function class $\mathcal{L}$ is called *online linear learnable* for short. The richer loss function class we consider is denoted by $\mathcal{C}$ and is a set of convex loss functions $\ell : \mathbb{R}^d \to \mathbb{R}$ satisfying some regularity conditions specified in terms of certain parameters described below.

We define a few parameters of the class $\mathcal{C}$. For any $b > 0$, let $\mathbb{B}^d(b) = \{\mathbf{y} \in \mathbb{R}^d : \|\mathbf{y}\| \le b\}$ be the ball of radius $b$. The class $\mathcal{C}$ is said to have Lipschitz constant $L_b$ on $\mathbb{B}^d(b)$ if for all $\ell \in \mathcal{C}$ and all $\mathbf{y} \in \mathbb{B}^d(b)$ there is an efficiently computable subgradient $\nabla \ell(\mathbf{y})$ with norm at most $L_b$. Next, $\mathcal{C}$ is said to be $\beta_b$-smooth on $\mathbb{B}^d(b)$ if for all $\ell \in \mathcal{C}$ and all $\mathbf{y}, \mathbf{y}' \in \mathbb{B}^d(b)$ we have

$$\ell(\mathbf{y}') \ \le \ \ell(\mathbf{y}) + \nabla \ell(\mathbf{y}) \cdot (\mathbf{y}' - \mathbf{y}) + \frac{\beta_b}{2}\|\mathbf{y} - \mathbf{y}'\|^2.$$

Next, define the projection operator $\Pi_b : \mathbb{R}^d \to \mathbb{B}^d(b)$ as $\Pi_b(\mathbf{y}) := \arg\min_{\mathbf{y}' \in \mathbb{B}^d(b)} \|\mathbf{y} - \mathbf{y}'\|$, and define $\epsilon_b := \sup_{\mathbf{y} \in \mathbb{R}^d, \ \ell \in \mathcal{C}} \frac{\ell(\Pi_b(\mathbf{y})) - \ell(\mathbf{y})}{\|\Pi_b(\mathbf{y}) - \mathbf{y}\|}$.

## 3  Online Boosting Algorithms

The setup is that we are given a $D$-bounded reference class of functions $\mathcal{F}$ with an online linear learning algorithm $\mathcal{A}$ with regret bound $R(\cdot)$. For normalization, we also assume that the output of $\mathcal{A}$ at any time is bounded in norm by $D$, i.e. $\|\mathcal{A}(\mathbf{x}_t)\| \le D$ for all $t$. We further assume that for every $b > 0$, we can compute[3] a Lipschitz constant $L_b$, a smoothness parameter $\beta_b$, and the parameter $\epsilon_b$ for the class $\mathcal{C}$ over $\mathbb{B}^d(b)$. Furthermore, the online boosting algorithm may make up to $N$ calls per iteration to any copies of $\mathcal{A}$ it maintains, for a given a budget parameter $N$.

Given this setup, our main result is an online boosting algorithm, Algorithm 1, competing with $\mathrm{span}(\mathcal{F})$. The algorithm maintains $N$ copies of $\mathcal{A}$, denoted $\mathcal{A}^i$, for $i = 1, 2, \ldots, N$. Each copy corresponds to one stage in boosting. When it receives a new example $\mathbf{x}_t$, it passes it to each $\mathcal{A}^i$ and obtains their predictions $\mathcal{A}^i(\mathbf{x}_t)$, which it then combines into a prediction for $\mathbf{y}_t$ using a linear combination. At the most basic level, this linear combination is simply the sum of all the predictions scaled by a step size parameter $\eta$. Two tweaks are made to this sum in step 8 to facilitate the analysis:

1. While constructing the sum, the partial sum $\mathbf{y}_t^{i-1}$ is multiplied by a *shrinkage* factor $(1 - \sigma_t^i \eta)$. This shrinkage term is tuned using an online gradient descent algorithm in step 14. The goal of the tuning is to induce the partial sums $\mathbf{y}_t^{i-1}$ to be aligned with a descent direction for the loss functions, as measured by the inner product $\nabla \ell_t(\mathbf{y}_t^{i-1}) \cdot y_t^{i-1}$.

2. The partial sums $\mathbf{y}_t^i$ are made to lie in $\mathbb{B}^d(B)$, for some parameter $B$, by using the projection operator $\Pi_B$. This is done to ensure that the Lipschitz constant and smoothness of the loss function are suitably bounded.

**Algorithm 1** Online Gradient Boosting for span($\mathcal{F}$)
---
**Require:** Number of weak learners $N$, step size parameter $\eta \in [\frac{1}{N}, 1]$,
 1: Let $B = \min\{\eta ND,\ \inf\{b \geq D : \eta \beta_b b^2 \geq \epsilon_b D\}\}$.
 2: Maintain $N$ copies of the algorithm $\mathcal{A}$, denoted $\mathcal{A}^i$ for $i = 1, 2, \ldots, N$.
 3: For each $i$, initialize $\sigma_1^i = 0$.
 4: **for** $t = 1$ **to** $T$ **do**
 5:     Receive example $\mathbf{x}_t$.
 6:     Define $\mathbf{y}_t^0 = \mathbf{0}$.
 7:     **for** $i = 1$ **to** $N$ **do**
 8:         Define $\mathbf{y}_t^i = \Pi_B((1 - \sigma_t^i \eta)\mathbf{y}_t^{i-1} + \eta \mathcal{A}^i(\mathbf{x}_t))$.
 9:     **end for**
10:     Predict $\mathbf{y}_t = \mathbf{y}_t^N$.
11:     Obtain loss function $\ell_t$ and suffer loss $\ell_t(\mathbf{y}_t)$.
12:     **for** $i = 1$ **to** $N$ **do**
13:         Pass loss function $\ell_t^i(\mathbf{y}) = \frac{1}{L_B} \nabla \ell_t(\mathbf{y}_t^{i-1}) \cdot \mathbf{y}$ to $\mathcal{A}^i$.
14:         Set $\sigma_{t+1}^i = \max\{\min\{\sigma_t^i + \alpha_t \nabla \ell_t(\mathbf{y}_t^{i-1}) \cdot \mathbf{y}_t^{i-1}), 1\}, 0\}$, where $\alpha_t = \frac{1}{L_B B \sqrt{t}}$.
15:     **end for**
16: **end for**
---

Once the boosting algorithm makes the prediction $\mathbf{y}_t$ and obtains the loss function $\ell_t$, each $\mathcal{A}^i$ is updated using a suitably scaled linear approximation to the loss function at the partial sum $\mathbf{y}_t^{i-1}$, i.e. the linear loss function $\frac{1}{L_B} \nabla \ell_t(\mathbf{y}_t^{i-1}) \cdot \mathbf{y}$. This forces $\mathcal{A}^i$ to produce predictions that are aligned with a descent direction for the loss function.

For lack of space, we provide the analysis of the algorithm in Section B in the supplementary material. The analysis yields the following regret bound for the algorithm:

**Theorem 1.** *Let $\eta \in [\frac{1}{N}, 1]$ be a given parameter. Let $B = \min\{\eta ND,\ \inf\{b \geq D : \eta \beta_b b^2 \geq \epsilon_b D\}\}$. Algorithm 1 is an online learning algorithm for span($\mathcal{F}$) and losses in $\mathcal{C}$ with the following regret bound for any $f \in \text{span}(\mathcal{F})$:*

$$R_f'(T) \leq \left(1 - \frac{\eta}{\|f\|_1}\right)^N \Delta_0 + 3\eta \beta_B B^2 \|f\|_1 T + L_B \|f\|_1 R(T) + 2L_B B \|f\|_1 \sqrt{T},$$

*where $\Delta_0 := \sum_{t=1}^T \ell_t(\mathbf{0}) - \ell_t(f(\mathbf{x}_t))$.*

The regret bound in this theorem depends on several parameters such as $B$, $\beta_B$ and $L_B$. In applications of the algorithm for 1-dimensional regression with commonly used loss functions, however, these parameters are essentially modest constants; see Section 3.1 for calculations of the parameters for various loss functions. Furthermore, if $\eta$ is appropriately set (e.g. $\eta = (\log N)/N$), then the average regret $R_f'(T)/T$ clearly converges to 0 as $N \to \infty$ and $T \to \infty$. While the requirement that $N \to \infty$ may raise concerns about computational efficiency, this is in fact analogous to the guarantee in the batch setting: the algorithms converge only when the number of boosting stages goes to infinity. Moreover, our lower bound (Theorem 4) shows that this is indeed necessary.

We also present a simpler boosting algorithm, Algorithm 2, that competes with CH($\mathcal{F}$). Algorithm 2 is similar to Algorithm 1, with some simplifications: the final prediction is simply a convex combination of the predictions of the base learners, with no projections or shrinkage necessary. While Algorithm 1 is more general, Algorithm 2 may still be useful in practice when a bound on the norm of the comparator function is known in advance, using the observations in Section 4.2. Furthermore, its analysis is cleaner and easier to understand for readers who are familiar with the Frank-Wolfe method, and this serves as a foundation for the analysis of Algorithm 1. This algorithm has an optimal (up to constant factors) regret bound as given in the following theorem, proved in Section A in the supplementary material. The upper bound in this theorem is proved along the lines of the Frank-Wolfe [8] algorithm, and the lower bound using information-theoretic arguments.

**Theorem 2.** *Algorithm 2 is an online learning algorithm for $CH(\mathcal{F})$ for losses in $\mathcal{C}$ with the regret bound*

$$R'(T) \leq \frac{8\beta_D D^2}{N}T + L_D R(T).$$

*Furthermore, the dependence of this regret bound on $N$ is optimal up to constant factors.*

The dependence of the regret bound on $R(T)$ is unimprovable without additional assumptions: otherwise, Algorithm 2 will be an online linear learning algorithm over $\mathcal{F}$ with better than $R(T)$ regret.

---

**Algorithm 2** Online Gradient Boosting for $CH(\mathcal{F})$

---

1: Maintain $N$ copies of the algorithm $\mathcal{A}$, denoted $\mathcal{A}^1, \mathcal{A}^2, \ldots, \mathcal{A}^N$, and let $\eta_i = \frac{2}{i+1}$ for $i = 1, 2, \ldots, N$.
2: **for** $t = 1$ **to** $T$ **do**
3:   Receive example $\mathbf{x}_t$.
4:   Define $\mathbf{y}_t^0 = \mathbf{0}$.
5:   **for** $i = 1$ **to** $N$ **do**
6:     Define $\mathbf{y}_t^i = (1 - \eta_i)\mathbf{y}_t^{i-1} + \eta_i \mathcal{A}^i(\mathbf{x}_t)$.
7:   **end for**
8:   Predict $\mathbf{y}_t = \mathbf{y}_t^N$.
9:   Obtain loss function $\ell_t$ and suffer loss $\ell_t(\mathbf{y}_t)$.
10:   **for** $i = 1$ **to** $N$ **do**
11:     Pass loss function $\ell_t^i(\mathbf{y}) = \frac{1}{L_D}\nabla\ell_t(\mathbf{y}_t^{i-1}) \cdot \mathbf{y}$ to $\mathcal{A}^i$.
12:   **end for**
13: **end for**

---

**Using a deterministic base online linear learning algorithm.** If the base online linear learning algorithm $\mathcal{A}$ is deterministic, then our results can be improved, because our online boosting algorithms are also deterministic, and using a standard simple reduction, we can now allow $\mathcal{C}$ to be any set of convex functions (smooth or not) with a computable Lipschitz constant $L_b$ over the domain $\mathbb{B}^d(b)$ for any $b > 0$.

This reduction converts arbitrary convex loss functions into linear functions: viz. if $\mathbf{y}_t$ is the output of the online boosting algorithm, then the loss function provided to the boosting algorithm as feedback is the linear function $\ell_t'(\mathbf{y}) = \nabla\ell_t(\mathbf{y}_t) \cdot \mathbf{y}$. This reduction immediately implies that the base online linear learning algorithm $\mathcal{A}$, when fed loss functions $\frac{1}{L_D}\ell_t'$, is already an online learning algorithm for $CH(\mathcal{F})$ with losses in $\mathcal{C}$ with the regret bound $R'(T) \leq L_D R(T)$.

As for competing with $\text{span}(\mathcal{F})$, since linear loss functions are 0-smooth, we obtain the following easy corollary of Theorem 1:

**Corollary 1.** *Let $\eta \in [\frac{1}{N}, 1]$ be a given parameter, and set $B = \eta ND$. Algorithm 1 is an online learning algorithm for $\text{span}(\mathcal{F})$ for losses in $\mathcal{C}$ with the following regret bound for any $f \in \text{span}(\mathcal{F})$:*

$$R_f'(T) \leq \left(1 - \frac{\eta}{\|f\|_1}\right)^N \Delta_0 + L_B\|f\|_1 R(T) + 2L_B B\|f\|_1\sqrt{T},$$

*where $\Delta_0 := \sum_{t=1}^T \ell_t(\mathbf{0}) - \ell_t(f(\mathbf{x}_t))$.*

## 3.1 The parameters for several basic loss functions

In this section we consider the application of our results to 1-dimensional regression, where we assume, for normalization, that the true labels of the examples and the predictions of the functions in the class $\mathcal{F}$ are in $[-1, 1]$. In this case $\|\cdot\|$ denotes the absolute value norm. Thus, in each round, the adversary chooses a labeled data point $(\mathbf{x}_t, y_t^\star) \in \mathcal{X} \times [-1, 1]$, and the loss for the prediction $y_t \in [-1, 1]$ is given by $\ell_t(y_t) = \ell(y_t^\star, y_t)$ where $\ell(\cdot, \cdot)$ is a fixed loss function that is convex in the second argument. Note that $D = 1$ in this setting. We

give examples of several such loss functions below, and compute the parameters $L_b$, $\beta_b$ and $\epsilon_b$ for every $b > 0$, as well as $B$ from Theorem 1.

1. Linear loss: $\ell(y^\star, y) = -y^\star y$. We have $L_b = 1$, $\beta_b = 0$, $\epsilon_b = 1$, and $B = \eta N$.
2. $p$-norm loss, for some $p \geq 2$: $\ell(y^\star, y) = |y^\star - y|^p$. We have $L_b = p(b+1)^{p-1}$, $\beta_b = p(p-1)(b+1)^{p-2}$, $\epsilon_b = \max\{p(1-b)^{p-1}, 0\}$, and $B = 1$.
3. Modified least squares: $\ell(y^\star, y) = \frac{1}{2}\max\{1 - y^\star y, 0\}^2$. We have $L_b = b+1$, $\beta_b = 1$, $\epsilon_b = \max\{1 - b, 0\}$, and $B = 1$.
4. Logistic loss: $\ell(y^\star, y) = \ln(1 + \exp(-y^\star y))$. We have $L_b = \frac{\exp(b)}{1+\exp(b)}$, $\beta_b = \frac{1}{4}$, $\epsilon_b = \frac{\exp(-b)}{1+\exp(-b)}$, and $B = \min\{\eta N, \ln(4/\eta)\}$.

# 4 Variants of the boosting algorithms

Our boosting algorithms and the analysis are considerably flexible: it is easy to modify the algorithms to work with a different (and perhaps more natural) kind of base learner which does greedy fitting, or incorporate a scaling of the base functions which improves performance. Also, when specialized to the batch setting, our algorithms provide better convergence rates than previous work.

## 4.1 Fitting to actual loss functions

The choice of an online *linear* learning algorithm over the base function class in our algorithms was made to ease the analysis. In practice, it is more common to have an online algorithm which produce predictions with comparable accuracy to the best function in hindsight for the *actual* sequence of loss functions. In particular, a common heuristic in boosting algorithms such as the original gradient boosting algorithm by Friedman [10] or the matching pursuit algorithm of Mallat and Zhang [18] is to build a linear combination of base functions by iteratively augmenting the current linear combination via greedily choosing a base function and a step size for it that minimizes the loss with respect to the residual label. Indeed, the boosting algorithm of Zhang and Yu [24] also uses this kind of greedy fitting algorithm as the base learner.

In the online setting, we can model greedy fitting as follows. We first fix a step size $\alpha \geq 0$ in advance. Then, in each round $t$, the base learner $\mathcal{A}$ receives not only the example $\mathbf{x}_t$, but also an *offset* $\mathbf{y}_t' \in \mathbb{R}^d$ for the prediction, and produces a prediction $\mathcal{A}(\mathbf{x}_t) \in \mathbb{R}^d$, after which it receives the loss function $\ell_t$ and suffers loss $\ell_t(\mathbf{y}_t' + \alpha\mathcal{A}(\mathbf{x}_t))$. The predictions of $\mathcal{A}$ satisfy

$$\sum_{t=1}^{T} \ell_t(\mathbf{y}_t' + \alpha\mathcal{A}(\mathbf{x}_t)) \leq \inf_{f \in \mathcal{F}} \sum_{t=1}^{T} \ell_t(\mathbf{y}_t' + \alpha f(\mathbf{x}_t)) + R(T),$$

where $R$ is the regret. Our algorithms can be made to work with this kind of base learner as well. The details can be found in Section C.1 of the supplementary material.

## 4.2 Improving the regret bound via scaling

Given an online linear learning algorithm $\mathcal{A}$ over the function class $\mathcal{F}$ with regret $R$, then for any scaling parameter $\lambda > 0$, we trivially obtain an online linear learning algorithm, denoted $\lambda\mathcal{A}$, over a $\lambda$-scaling of $\mathcal{F}$, viz. $\lambda\mathcal{F} := \{\lambda f \mid f \in \mathcal{F}\}$, simply by multiplying the predictions of $\mathcal{A}$ by $\lambda$. The corresponding regret scales by $\lambda$ as well, i.e. it becomes $\lambda R$.

The performance of Algorithm 1 can be improved by using such an online linear learning algorithm over $\lambda\mathcal{F}$ for a suitably chosen scaling $\lambda \geq 1$ of the function class $\mathcal{F}$. The regret bound from Theorem 1 improves because the 1-norm of $f$ measured with respect to $\lambda\mathcal{F}$, i.e. $\|f\|_1' = \max\{1, \frac{\|f\|_1}{\lambda}\}$, is smaller than $\|f\|_1$, but degrades because the parameter $B' = \min\{\eta N\lambda D, \inf\{b \geq \lambda D : \eta\beta_b b^2 \geq \epsilon_b\lambda D\}\}$ is larger than $B$. But, as detailed in Section C.2 of the supplementary material, in many situations the improvement due to the former compensates for the degradation due to the latter, and overall we can get improved regret bounds using a suitable value of $\lambda$.

### 4.3 Improvements for batch boosting

Our algorithmic technique can be easily specialized and modified to the standard batch setting with a fixed batch of training examples and a base learning algorithm operating over the batch, exactly as in [24]. The main difference compared to the algorithm of [24] is the use of the $\sigma$ variables to scale the coefficients of the weak hypotheses appropriately. While a seemingly innocuous tweak, this allows us to derive analogous bounds to those of Zhang and Yu [24] on the optimization error that show that our boosting algorithm converges exponential faster. A detailed comparison can be found in Section C.3 of the supplementary material.

## 5 Experimental Results

Is it possible to boost in an online fashion in practice with real base learners? To study this question, we implemented and evaluated Algorithms 1 and 2 within the Vowpal Wabbit (VW) open source machine learning system [23]. The three online base learners used were VW's default linear learner (a variant of stochastic gradient descent), two-layer sigmoidal neural networks with 10 hidden units, and regression stumps.

Regression stumps were implemented by doing stochastic gradient descent on each individual feature, and predicting with the best-performing non-zero valued feature in the current example.

All experiments were done on a collection of 14 publically available regression and classification datasets (described in Section D in the supplementary material) using squared loss. The only parameters tuned were the learning rate and the number of weak learners, as well as the step size parameter for Algorithm 1. Parameters were tuned based on progressive validation loss on half of the dataset; reported is propressive validation loss on the remaining half. Progressive validation is a standard online validation technique, where each training example is used for testing before it is used for updating the model [3].

The following table reports the average and the median, over the datasets, relative improvement in squared loss over the respective base learner. Detailed results can be found in Section D in the supplementary material.

| Base learner | Average relative improvement | | Median relative improvement | |
|---|---|---|---|---|
| | Algorithm 1 | Algorithm 2 | Algorithm 1 | Algorithm 2 |
| SGD | 1.65% | 1.33% | 0.03% | 0.29% |
| Regression stumps | 20.22% | 15.9% | 10.45% | 13.69% |
| Neural networks | 7.88% | 0.72% | 0.72% | 0.33% |

Note that both SGD (stochastic gradient descent) and neural networks are already very strong learners. Naturally, boosting is much more effective for regression stumps, which is a weak base learner.

## 6 Conclusions and Future Work

In this paper we generalized the theory of boosting for regression problems to the online setting and provided online boosting algorithms with theoretical convergence guarantees. Our algorithmic technique also improves convergence guarantees for batch boosting algorithms. We also provide experimental evidence that our boosting algorithms do improve prediction accuracy over commonly used base learners in practice, with greater improvements for weaker base learners. The main remaining open question is whether the boosting algorithm for competing with the span of the base functions is optimal in any sense, similar to our proof of optimality for the the boosting algorithm for competing with the convex hull of the base functions.

## Footnotes

[1] There is a slight abuse of notation here. $\mathcal{A}(\cdot)$ is not a function but rather the output of the online learning algorithm $\mathcal{A}$ computed on the given example using its internal state.

[2] This is without loss of generality; as will be seen momentarily, our base assumption only requires an online learning algorithm $\mathcal{A}$ for $\mathcal{F}$ for linear losses $\ell_t$. By running the Hedge algorithm on two copies of $\mathcal{A}$, one of which receives the actual loss functions $\ell_t$ and the other recieves $-\ell_t$, we get an algorithm which competes with negations of functions in $\mathcal{F}$ and the constant zero function as well. Furthermore, since the loss functions are convex (indeed, linear) this can be made into a deterministic reduction by choosing the convex combination of the outputs of the two copies of $\mathcal{A}$ with mixing weights given by the Hedge algorithm.

[3]It suffices to compute upper bounds on these parameters.

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
