[Supplementary Material]

# Supplementary material for "Online Gradient Boosting"

## A    Competing with convex combinations of the base functions

We give the analysis of Algorithm 2 before that of Algorithm 1 since it is easier to understand and provides the foundation for the analysis of Algorithm 1. In this section we show that Algorithm 2 satisfies the regret bound claimed in Theorem 2, restated here for convenience.

**Theorem 3.** *Algorithm 2 is an online learning algorithm for $CH(\mathcal{F})$ for losses in $\mathcal{C}$ with the regret bound*

$$R'(T) \ \leq \ \frac{8\beta_D D^2}{N}T + L_D R(T).$$

*Proof.* First, note that for any $i = 1, 2, \ldots, N$, since $\ell_t^i$ is a linear function, we have

$$\inf_{f \in \mathrm{CH}(\mathcal{F})} \sum_{t=1}^{T} \ell_t^i(f(\mathbf{x}_t)) \ = \ \inf_{f \in \mathcal{F}} \sum_{t=1}^{T} \ell_t^i(f(\mathbf{x}_t)).$$

Let $f$ be any function in $\mathrm{CH}(\mathcal{F})$. The equality above and the fact that $\mathcal{A}^i$ is an online learning algorithm for $\mathcal{F}$ with regret bound $R(\cdot)$ for the 1-Lipschitz linear loss functions $\ell_t^i(\mathbf{y}) = \frac{1}{L_D}\nabla \ell_t(\mathbf{y}_t^{i-1}) \cdot \mathbf{y}$ imply that

$$\sum_{t=1}^{T} \frac{1}{L_D}\nabla \ell_t(\mathbf{y}_t^{i-1}) \cdot \mathcal{A}^i(\mathbf{x}_t) \ \leq \ \sum_{t=1}^{T} \frac{1}{L_D}\nabla \ell_t(\mathbf{y}_t^{i-1}) \cdot f(\mathbf{x}_t) + R(T). \qquad (2)$$

Now define, for $i = 0, 1, 2, \ldots, N$, $\Delta_i = \sum_{t=1}^{T} \ell_t(\mathbf{y}_t^i) - \ell_t(f(\mathbf{x}_t))$. We have

$$\Delta_i \ = \ \sum_{t=1}^{T} \ell_t(\mathbf{y}_t^{i-1} + \eta_i(\mathcal{A}^i(\mathbf{x}_t) - \mathbf{y}_t^{i-1})) - \ell_t(f(\mathbf{x}_t))$$

$$\leq \ \sum_{t=1}^{T} \ell_t(\mathbf{y}_t^{i-1}) - \ell_t(f(\mathbf{x}_t)) + \eta_i \nabla \ell_t(\mathbf{y}_t^{i-1}) \cdot (\mathcal{A}^i(\mathbf{x}_t) - \mathbf{y}_t^{i-1}) + \frac{\eta_i^2 \beta_D}{2}\|\mathcal{A}^i(\mathbf{x}_t) - \mathbf{y}_t^{i-1}\|^2$$

(by $\beta_D$-smoothness of $\mathcal{C}$)

$$\leq \ \left[\sum_{t=1}^{T} \ell_t(\mathbf{y}_t^{i-1}) - \ell_t(f(\mathbf{x}_t)) + \eta_i \nabla \ell_t(\mathbf{y}_t^{i-1}) \cdot (f(\mathbf{x}_t) - \mathbf{y}_t^{i-1}) + 2\eta_i^2 \beta_D D^2\right] + \eta_i L_D R(T)$$

(by (2) and using the bound $\|\mathcal{A}^i(\mathbf{x}_t) - \mathbf{y}_t^{i-1}\| \leq 2D$)

$$\leq \ \left[\sum_{t=1}^{T} \ell_t(\mathbf{y}_t^{i-1}) - \ell_t(f(\mathbf{x}_t)) - \eta_i(\ell_t(\mathbf{y}_t^{i-1}) - \ell_t(f(\mathbf{x}_t))) + 2\eta_i^2 \beta_D D^2\right] + \eta_i L_D R(T)$$

(by convexity, $\ell_t(\mathbf{y}_t^{i-1}) + \nabla \ell(\mathbf{y}_t^{i-1}) \cdot (f(\mathbf{x}_t) - \mathbf{y}_t^{i-1}) \leq \ell_t(f(\mathbf{x}_t))$)

$$\leq \ (1 - \eta_i)\Delta_{i-1} + 2\eta_i^2 \beta_D D^2 T + \eta_i L_D R(T).$$

For $i = 1$, since $\eta_1 = 1$, the above bound implies that $\Delta_1 \leq 2\beta_D D^2 T + L_D R(T)$. Starting from this base case, an easy induction on $i \geq 1$ proves that $\Delta_i \leq \frac{8\beta_D D^2}{i}T + L_D R(T)$. Applying this bound for $i = N$ completes the proof. $\qquad\square$

We now show that the dependence of the regret bound of Algorithm 2 on the parameter $N$ is optimal up to constant factors.

**Theorem 4.** *Let $N$ be any specified bound on the total number of calls in each iteration to all copies of the base online linear learning algorithm. Then there is a setting of 1-dimensional prediction with a 1-bounded comparator function class $\mathcal{F}$, an online linear optimization algorithm $\mathcal{A}$ over $\mathcal{F}$, and a class $\mathcal{C}$ of loss functions that is 1-smooth on $\mathbb{R}$ such that any online boosting algorithm for $CH(\mathcal{F})$ with losses in $\mathcal{C}$ respecting the bound $N$ has regret at least $\Omega(\frac{T}{N})$.*

*Proof.* Consider the following construction. At a high level, the setting is 1-dimensional regression with $\mathcal{C}$ corresponding to squared loss. The domain $\mathcal{X} = \mathbb{N}$ and true labels of examples are in $[0, 1]$.

Define $p_1 = \frac{1}{2} + \epsilon$ and $p_2 = \frac{1}{2} - \epsilon$, where $\epsilon = \frac{1}{10\sqrt{N}}$, and let $D_1$ and $D_2$ be two distributions over $\{0, 1\}^N$ where each bit is Bernoulli random variable with parameter $p_1$ and $p_2$ respectively, chosen independently of the other bits. Consider a sequence of examples $(\mathbf{x}_t, y_t^\star) \in \mathbb{N} \times [0, 1]$ generated as follows: $\mathbf{x}_t = t$, and the label $y_t^\star$ is chosen from $\{p_1, p_2\}$ uniformly at random in each round.

Let for $c = \frac{1}{4000}$. The function class $\mathcal{F}$ consists of a large number, $M = \frac{1}{c}N$, of functions $f_i$, $i \in [M]$. For each $t$ and $i$, we set $f_i(\mathbf{x}_t) = 1$ w.p. $y_t^\star$, and $0$ w.p. $1 - y_t^\star$, independently of all other values of $t$ and $i$.

The base online linear learning algorithm $\mathcal{A}$ is simply Hedge over the $M$ functions. In each round, the Hedge algorithm selects one of the $M$ functions in $\mathcal{F}$ and uses that to predict the label, and for any sequence of $T$ examples, with high probability, incurs regret $R(T) = O(\sqrt{\log(M)T})$.

We set $\mathcal{C}$ to be set of squared loss functions, i.e. functions of the form $\ell(y) = \frac{1}{2}(y - y^\star)^2$ for $y^\star \in [0, 1]$. Note that these loss functions are 1-smooth and $D = 1$. In round $t$, the loss function is $\ell_t(y) = \frac{1}{2}(y - y_t^\star)^2$.

Consider the function $\bar{f} = \frac{1}{M} \sum_{i \in [M]} f_i$, which is in $\mathrm{CH}(\mathcal{F})$. Given any input sequence $(\mathbf{x}_t, y_t^\star)$ for $t = 1, 2, \ldots, T$ it is easy to calculate that $\mathbb{E}[\frac{1}{2}(\bar{f}(\mathbf{x}_t) - y_t^\star)^2] = \frac{y_t^\star(1 - y_t^\star)}{2M} \leq \frac{1}{2M}$, and since the examples and predictions of functions on the examples are independent across iterations, a simple application of the multiplicative Chernoff bound implies that if $T \geq 12M$, then with probability at least 0.9, we have $\sum_{t=1}^T \frac{1}{2}(\bar{f}(\mathbf{x}_t) - y_t^\star)^2 \leq \frac{T}{M}$.

Now suppose there is an online boosting algorithm making at most $N$ calls total to all copies of $\mathcal{A}$ in each iteration, that for any large enough $T$ and for any sequence $(\mathbf{x}_t, y_t^\star)$ for $t = 1, 2, \ldots, T$, outputs predictions $y_t$ such that with high probability, say at least 0.9, we have $\sum_{t=1}^T \frac{1}{2}(y_t - y_t^\star)^2 \leq \sum_{t=1}^T \frac{1}{2}(\bar{f}(\mathbf{x}_t) - y_t^\star)^2 + \frac{cT}{N}$. Then by a union bound, with probability at least 0.8, we have $\sum_{t=1}^T \frac{1}{2}(y_t - y_t^\star)^2 \leq \frac{cT}{N} + \frac{T}{M} \leq \frac{2cT}{N}$. By Markov's inequality and a union bound, with probability at least 0.7, for a uniform random time $\tau \in [T]$, we have

$$\frac{1}{2}(y_\tau - y_\tau^\star)^2 \leq \frac{20c}{N} = \frac{\epsilon^2}{2}, \tag{3}$$

or in other words, $y_\tau$ is on the same side of $\frac{1}{2}$ as $y_\tau^\star$, and thus can be used to identify $y_\tau^\star$. In the rest of the proof, we will use this fact, along with fact the total variation distance between $D_1$ and $D_2$, denoted $d_{\mathrm{TV}}(D_1, D_2)$, is small, to derive a contradiction.

Define the random variable $Y : \{0, 1\}^N \to \mathbb{R}$ as follows. For any bit string $s = \langle s_1, s_2, \ldots, s_N \rangle \in \{0, 1\}^N$, choose a random round $\tau \in [T]$, and simulate the online boosting process until round $\tau - 1$ by sampling $y_t^\star$'s and the outputs of $f_i(\mathbf{x}_t)$ for all $t \leq \tau - 1$ and $i \in [M]$ from the appropriate distributions. In round $\tau$, let $f_{i_1}, f_{i_2}, \ldots, f_{i_N}$ be the functions that are obtained from the at most $N$ calls to copies of $\mathcal{A}$ (there could be repetitions). Assign $f_{i_j}(\mathbf{x}_\tau) = s_j$ for $j \in [N]$ (being careful with repeated functions and repeating outputs appropriately), and run the booster with these outputs to obtain $y_\tau$, and set $Y(s) = y_\tau$. Let $\Pr[\cdot]$ denotes probability of events in this process for generating $Y(s)$ given $s$.

Let $\mathbb{E}_1[X(s)]$ and $\mathbb{E}_2[X(s)]$ denote expectation of a random variable $X : \{0, 1\}^N \to \mathbb{R}$ when $s$ is drawn from $D_1$ and $D_2$ respectively, and let $\mathbb{E}_0[X(I, s)]$ denote expectation of a random variable $X : \{1, 2\} \times \{0, 1\}^N \to \mathbb{R}$ when $I$ is chosen from $\{1, 2\}$ uniformly at random and then $s$ is sampled from $D_I$. The above analysis (inequality (3)) implies that

$$0.7 \leq \mathbb{E}_0[\Pr[|Y(s) - p_I| \leq \epsilon]] = \frac{1}{2}\mathbb{E}_1[\Pr[|Y(s) - p_1| \leq \epsilon]] + \frac{1}{2}\mathbb{E}_2[\Pr[|Y(s) - p_2| \leq \epsilon]].$$

Now define a random variable $X : \{0, 1\}^N \to \mathbb{R}$ as $X(s) = \Pr[Y(s) \geq \frac{1}{2}]$. Since

$$\Pr[Y(s) \geq \tfrac{1}{2}] \geq \Pr[|Y(s) - p_1| \leq \epsilon] \quad \text{and} \quad 1 - \Pr[Y(s) \geq \tfrac{1}{2}] \geq \Pr[|Y(s) - p_2| \leq \epsilon],$$

we conclude, using the above bound, that $\mathbb{E}_1[X(s)] - \mathbb{E}_2[X(s)] \geq 0.4$. This is a contradiction, since because $X(s) \in [0,1]$, we have

$$\mathbb{E}_1[X(s)] - \mathbb{E}_2[X(s)] \ \leq \ d_{\mathrm{TV}}(D_1, D_2) \ < \ 4\sqrt{\epsilon^2 N} \ = \ 0.4,$$

where the bound on $d_{\mathrm{TV}}(D_1, D_2)$ is standard, for e.g. see [15]. This gives us the desired contradiction. $\qquad\square$

The above result can be easily extended to any given parameters $\beta$ and $D$ so that the $\mathcal{F}$ is $D$-bounded and $\mathcal{C}$ is $\beta$-smooth on $\mathbb{R}$, giving a lower bound of $\Omega(\frac{\beta D^2 T}{N})$ on the regret of an online boosting algorithm for $\mathrm{CH}(\mathcal{F})$ with losses in $\mathcal{C}$: we simply scale all function and label values by $D$, and consider the loss functions $\ell(y, y^\star) = \frac{\beta}{2}(y - y^\star)^2$. If there were an online boosting algorithm for $\mathrm{CH}(\mathcal{F})$ with these loss functions with regret $o(\frac{\beta D^2 T}{N})$, then by scaling down the predictions by $D$, we obtain an online boosting algorithm for exactly the setting in the proof of Theorem 4 with a regret bound of $o(\frac{T}{N})$, which is a contradiction.

## B  Competing with the span of the base functions

In this section we show that Algorithm 1 satisfies the regret bound claimed in Theorem 1, restated here for convenience.

**Theorem 5.** *Let $\eta \in [\frac{1}{N}, 1]$ be a given parameter. Let $B = \min\{\eta N D,\ \inf\{b \geq D : \eta \beta_b b^2 \geq \epsilon_b D\}\}$. Algorithm 1 is an online learning algorithm for $\mathrm{span}(\mathcal{F})$ and losses in $\mathcal{C}$ with the following regret bound for any $f \in \mathrm{span}(\mathcal{F})$:*

$$R'_f(T) \ \leq \ \left(1 - \frac{\eta}{\|f\|_1}\right)^N \Delta_0 + 3\eta\beta_B B^2 \|f\|_1 T + L_B \|f\|_1 R(T) + 2L_B B \|f\|_1 \sqrt{T},$$

*where $\Delta_0 := \sum_{t=1}^{T} \ell_t(\mathbf{0}) - \ell_t(f(\mathbf{x}_t))$.*

*Proof.* Let $f = \sum_{g \in S} w_g g$, for some finite subset $S$ of $\mathcal{F}$, where $w_g \in \mathbb{R}$. Since $\mathcal{F}$ is symmetric, we may assume that all $w_g \geq 0$, and let $W := \sum_g w_g$. Furthermore, we may assume that $\mathbf{0} \in S$ with weight $w_{\mathbf{0}} = \max\{1 - \sum_{g \in S,\ g \neq \mathbf{0}} w_g, 0\}$, so that $W \geq 1$. Note that $\|f\|_1$ is exactly the infimum of $W$ over all such ways of expressing $f$ as a finite weighted sum of functions in $\mathcal{F}$. We now prove that bound stated in the theorem holds with $\|f\|_1$ replaced by $W$; the theorem then follows simply by taking the infimum of the bound over all such ways of expressing $f$.

Now, for each $i \in [N]$, the update in line 14 of Algorithm 1 is exactly online gradient descent [25] on the domain $[0,1]$ with linear loss functions $\sigma \mapsto -\nabla\ell_t(\mathbf{y}_t^{i-1}) \cdot \mathbf{y}_t^{i-1}\sigma$. Note that the derivative of this loss function is bounded as follows: $|-\nabla\ell_t(\mathbf{y}_t^{i-1}) \cdot \mathbf{y}_t^{i-1}| \leq L_B B$. Since $\frac{1}{W} \in [0,1]$, the standard analysis of online gradient descent then implies that the sequence $\sigma_t^i$ for $t = 1, 2, \ldots, T$ satisfies

$$\sum_{t=1}^{T} -\nabla\ell_t(\mathbf{y}_t^{i-1}) \cdot \mathbf{y}_t^{i-1}\sigma_t^i \ \leq \ \sum_{t=1}^{T} -\nabla\ell_t(\mathbf{y}_t^{i-1}) \cdot \mathbf{y}_t^{i-1}\frac{1}{W} + 2L_B B\sqrt{T}. \tag{4}$$

Next, since $f = \sum_{g \in S} w_g g$ with $w_g \geq 0$, we have

$$\frac{1}{W}\sum_{t=1}^{T} \nabla\ell_t(\mathbf{y}_t^i) \cdot f(\mathbf{x}_t) = \frac{1}{\sum_{g \in S} w_g}\sum_{t=1}^{T}\sum_{g \in S} w_g \nabla\ell_t(\mathbf{y}_t^i) \cdot g(\mathbf{x}_t) \ \geq \ \min_{g \in S}\sum_{t=1}^{T} \nabla\ell_t(\mathbf{y}_t^i) \cdot g(\mathbf{x}_t). \tag{5}$$

Let $g^\star \in \arg\min_{g \in S}\sum_{t=1}^{T} \nabla\ell_t(\mathbf{y}_t^i) \cdot g(\mathbf{x}_t)$. Since $\mathcal{A}^i$ is an online learning algorithm for $\mathcal{F}$ with regret bound $R(\cdot)$ for the 1-Lipschitz linear loss functions $\ell_t^i(\mathbf{y}) = \frac{1}{L_B}\nabla\ell_t(\mathbf{y}_t^{i-1}) \cdot \mathbf{y}$, and $g^\star \in \mathcal{F}$, multiplying the regret bound (1) by $L_B$ we have

$$\sum_{t=1}^{T} \nabla\ell_t(\mathbf{y}_t^{i-1}) \cdot \mathcal{A}^i(\mathbf{x}_t) \leq \sum_{t=1}^{T} \nabla\ell_t(\mathbf{y}_t^{i-1}) \cdot g^\star(\mathbf{x}_t) + L_B R(T) \leq \frac{1}{W}\sum_{t=1}^{T} \nabla\ell_t(\mathbf{y}_t^{i-1}) \cdot f(\mathbf{x}_t) + L_B R(T)$$

$$\tag{6}$$

by (5). Now, we analyze how much excess loss is potentially introduced due to the projection in line 8. First, note that if $B = \eta ND$, then the projection has no effect since $(1 - \sigma_t^i \eta)\mathbf{y}_t^{i-1} + \eta\mathcal{A}^i(\mathbf{x}_t) \in \mathbb{B}^d(B)$, and in this case $\ell_t(\mathbf{y}_t^i) = \ell_t((1 - \sigma_t^i \eta)\mathbf{y}_t^{i-1} + \eta\mathcal{A}^i(\mathbf{x}_t))$. If $B < \eta ND$, then by the definition of $B$, $\eta\beta_B B^2 \geq \epsilon_B D$, and since $(1 - \sigma_t^i \eta)\mathbf{y}_t^{i-1} \in \mathbb{B}^d(B)$ and $\|\eta\mathcal{A}^i(\mathbf{x}_t))\| \leq \eta D$, and we have

$$\ell_t(\mathbf{y}_t^i) = \ell_t(\Pi_B((1 - \sigma_t^i \eta)\mathbf{y}_t^{i-1} + \eta\mathcal{A}^i(\mathbf{x}_t))) \leq \ell_t((1 - \sigma_t^i \eta)\mathbf{y}_t^{i-1} + \eta\mathcal{A}^i(\mathbf{x}_t)) + \eta\epsilon_B D.$$

In either case, we have

$$\ell_t(\mathbf{y}_t^i) \leq \ell_t((1 - \sigma_t^i \eta)\mathbf{y}_t^{i-1} + \eta\mathcal{A}^i(\mathbf{x}_t)) + \eta^2\beta_B B^2. \tag{7}$$

We now move to the main part of the analysis. Define for $i = 0, 1, 2, \ldots, N$, $\Delta_i := \sum_{t=1}^{T} \ell_t(\mathbf{y}_t^i) - \ell_t(f(\mathbf{x}_t))$. We have

$$\Delta_i \leq \left[\sum_{t=1}^{T} \ell_t((1 - \sigma_t^i \eta)\mathbf{y}_t^{i-1} + \eta\mathcal{A}^i(\mathbf{x}_t)) - \ell_t(f(\mathbf{x}_t))\right] + \eta^2\beta_B B^2 T$$

$$\leq \Delta_{i-1} + \left[\sum_{t=1}^{T} \eta\nabla\ell_t(\mathbf{y}_t^{i-1}) \cdot (\mathcal{A}^i(\mathbf{x}_t) - \sigma_t^i\mathbf{y}_t^{i-1}) + \frac{\beta_B\eta^2}{2}\|\mathcal{A}^i(\mathbf{x}_t) - \sigma_t^i\mathbf{y}_t^{i-1}\|^2\right] + \eta^2\beta_B B^2 T$$

(by $\beta_B$-smoothness)

$$\leq \Delta_{i-1} + \left[\sum_{t=1}^{T} \frac{\eta}{W}\nabla\ell_t(\mathbf{y}_t^{i-1}) \cdot (f(\mathbf{x}_t) - \mathbf{y}_t^{i-1})\right] + 3\eta^2\beta_B B^2 T + \eta L_B R(T) + 2\eta L_B B\sqrt{T}$$

(by (4), (6) and the fact that $\|\mathcal{A}^i(\mathbf{x}_t) - \sigma_t^i\mathbf{y}_t^{i-1}\| \leq D + B \leq 2B$)

$$\leq \left(1 - \frac{\eta}{W}\right)\Delta_{i-1} + 3\eta^2\beta_B B^2 T + \eta L_B R(T) + 2\eta L_B B\sqrt{T},$$

since, by convexity of $\ell_t$ we have $\ell_t(\mathbf{y}_t^{i-1}) + \nabla\ell(\mathbf{y}_t^{i-1}) \cdot (f(\mathbf{x}_t) - \mathbf{y}_t^{i-1}) \leq \ell_t(f(\mathbf{x}_t))$. Applying the above bound iteratively, we get

$$\Delta_N \leq \left(1 - \frac{\eta}{W}\right)^N\Delta_0 + \sum_{i=1}^{N}\left(1 - \frac{\eta}{W}\right)^{i-1} \cdot (3\eta^2\beta_B B^2 T + \eta L_B R(T) + 2\eta L_B B\sqrt{T})$$

$$\leq \left(1 - \frac{\eta}{W}\right)^N\Delta_0 + 3\eta\beta_B B^2 WT + L_B W R(T) + 2L_B BW\sqrt{T}.$$

This completes the proof. $\qquad\square$

## C   Variants of the boosting algorithms

In this section we provide the omitted details of two variants of our boosting algorithms: (a) a variant that work with a different kind of base learner which does greedy fitting, and (b) a variant that incorporates a scaling of the base functions to improves performance. We also show how our algorithmic technique can be used to improve the convergence speed for batch boosting.

### C.1   Fitting to actual loss functions

The choice of an online *linear* learning algorithm over the base function class in our algorithms was made to ease the analysis. In practice, it is more common to have an online algorithm which produce predictions with comparable accuracy to the best function in hindsight for the *actual* sequence of loss functions. In particular, a common heuristic in boosting algorithms such as the original gradient boosting algorithm by Friedman [10] or the matching pursuit algorithm of Mallat and Zhang [18] is to build a linear combination of base functions by iteratively augmenting the current linear combination by greedily choosing a base function and a step size for it that minimizes the loss with respect to the residual label. Indeed, the boosting algorithm of Zhang and Yu [24] also uses this kind of greedy fitting algorithm as the base learner.

In the online setting, we can model greedy fitting as follows. We first fix a step size $\alpha \geq 0$ in advance. Then, in each round $t$, the base learner $\mathcal{A}$ receives not only the example $\mathbf{x}_t$, but also an *offset* $\mathbf{y}'_t \in \mathbb{R}^d$ for the prediction, and produces a prediction $\mathcal{A}(\mathbf{x}_t) \in \mathbb{R}^d$, after which it receives the loss function $\ell_t$ and suffers loss $\ell_t(\mathbf{y}'_t + \alpha \mathcal{A}(\mathbf{x}_t))$. The predictions of $\mathcal{A}$ satisfy

$$\sum_{t=1}^{T} \ell_t(\mathbf{y}'_t + \alpha \mathcal{A}(\mathbf{x}_t)) \leq \inf_{f \in \mathcal{F}} \sum_{t=1}^{T} \ell_t(\mathbf{y}'_t + \alpha f(\mathbf{x}_t)) + R(T),$$

where $R$ is the regret. We now describe how our algorithms can be made to work with this kind of base learner as well.

Assume that for some known parameter $B > 0$, we have $\|\mathbf{y}'_t\| \leq B$, for all $t$. Let $B' = B + \alpha D$, and assume that the loss functions $\ell_t$ are $L_{B'}$ Lipschitz and $\beta_{B'}$ smooth on $\mathbb{B}^d(B')$. Then using the convexity and smoothness of the loss functions, we have $\ell_t(\mathbf{y}'_t + \alpha \mathcal{A}(\mathbf{x}_t)) \geq \ell_t(\mathbf{y}'_t) + \alpha \nabla \ell_t(\mathbf{y}'_t) \cdot \mathcal{A}(\mathbf{x}_t)$ and $\ell_t(\mathbf{y}'_t + \alpha f(\mathbf{x}_t)) \leq \ell_t(\mathbf{y}'_t) + \alpha \nabla \ell_t(\mathbf{y}'_t) \cdot f(\mathbf{x}_t) + \frac{\beta_{B'}\alpha^2}{2}\|f(\mathbf{x}_t)\|^2$. Plugging these bounds into the above regret bound we get, for any $f \in \mathcal{F}$,

$$\sum_{t=1}^{T} \nabla \ell_t(\mathbf{y}'_t) \cdot \mathcal{A}(\mathbf{x}_t) \leq \sum_{t=1}^{T} \left( \nabla \ell_t(\mathbf{y}'_t) \cdot f(\mathbf{x}_t) + \frac{\beta_{B'}}{2}\alpha\|f(\mathbf{x}_t)\|^2 \right) + \frac{1}{\alpha} R(T).$$

Since $\|f(\mathbf{x}_t)\| \leq D$, setting $\alpha = \sqrt{\frac{2R(T)}{\beta_{B'}D^2T}}$, we conclude that

$$\sum_{t=1}^{T} \nabla \ell_t(\mathbf{y}'_t) \cdot \mathcal{A}(\mathbf{x}_t) \leq \sum_{t=1}^{T} \nabla \ell_t(\mathbf{y}'_t) \cdot f(\mathbf{x}_t) + \sqrt{2\beta_{B'}D^2TR(T)}. \tag{8}$$

This regret bound is sublinear in $T$ if $R(T)$ is sublinear. We can obtain a better regret bound if we assume that $R(T)$ scales linearly with $\alpha$: this is a natural assumption since the functions $\ell_t(\mathbf{y}'_t + \alpha \mathbf{y})$ are $\alpha L_{B'}$ Lipschitz in the prediction $\mathbf{y}$. In this case, the regret bound $R(T) = \alpha R'(T)$ for some fixed $R' : \mathbb{N} \to \mathbb{R}_+$ indepedent of $\alpha$, and we can choose $\alpha = \frac{2R'(T)}{\beta_{B'}D^2T}$ so that

$$\sum_{t=1}^{T} \nabla \ell_t(\mathbf{y}'_t) \cdot \mathcal{A}(\mathbf{x}_t) \leq \sum_{t=1}^{T} \nabla \ell_t(\mathbf{y}'_t) \cdot f(\mathbf{x}_t) + 2R'(T). \tag{9}$$

Either the bound (8) or (9) suffices for the analysis of our boosting algorithms to go through: to use this kind of base learner $\mathcal{A}$, we again keep $N$ copies $\mathcal{A}^1, \mathcal{A}^2, \ldots, \mathcal{A}^N$ with a suitably chosen step size $\alpha$, and simply change line 11 of Algorithm 2 and line 13 of Algorithm 1 to pass the offset $\mathbf{y}'_t = \mathbf{y}^{i-1}_t$ to $\mathcal{A}^i$.

### C.2 Improving the regret bound via scaling

Given an online linear learning algorithm $\mathcal{A}$ over the function class $\mathcal{F}$ with regret $R$, then for any scaling parameter $\lambda > 0$, we trivially obtain an online linear learning algorithm, denoted $\lambda \mathcal{A}$, over a $\lambda$-scaling of $\mathcal{F}$, viz. $\lambda \mathcal{F} := \{\lambda f \mid f \in \mathcal{F}\}$, simply by multiplying the predictions of $\mathcal{A}$ by $\lambda$. The corresponding regret scales by $\lambda$ as well, i.e. it becomes $\lambda R$.

The performance of Algorithm 1 can be improved by using such an online linear learning algorithm over $\lambda \mathcal{F}$ for a suitably chosen scaling $\lambda \geq 1$ of the function class $\mathcal{F}$. Let $\|f\|'_1 = \max\{1, \frac{\|f\|_1}{\lambda}\}$ be the 1-norm of $f$ measured with respect to $\lambda \mathcal{F}$, and $B' = \min\{\eta N \lambda D, \inf\{b \geq \lambda D : \eta \beta_b b^2 \geq \epsilon_b \lambda D\}\}$. Then we immediately get the following corollary of Theorem 1:

**Corollary 2.** *For any $f \in span(\mathcal{F})$, let $\Delta_0 = \sum_{t=1}^{T} \ell_t(0) - \ell_t(f(\mathbf{x}_t))$. Algorithm 1, using $\lambda \mathcal{A}$ as the online linear algorithm over $\lambda \mathcal{F}$, is an online learning algorithm for $span(\mathcal{F})$ for losses in $\mathcal{C}$ with the following regret bound for any $f \in span(\mathcal{F})$:*

$$R'_f(T) \leq \left(1 - \frac{\eta}{\|f\|'_1}\right)^N \Delta_0 + 3\eta\beta_{B'}B'^2\|f\|'_1 T + L_{B'}\|f\|'_1 \lambda R(T) + 2L_{B'}B'\|f\|'_1\sqrt{T}.$$

Choosing large values of $\lambda$ implies that $\|f\|_1'$ can be significantly smaller than $\|f\|_1$. But $B'$ becomes bigger than $B$, and correspondingly, the parameters $\beta_{B'}$ and $L_{B'}$ become bigger than $\beta_B$ and $L_B$ respectively. Also, the (lower order) dependence on the regret term $R(T)$ increases by a factor of $\lambda$.

However, it turns out (see Section 3.1) that in several common applications of the algorithm, $B'$ can be set to be equal to $B$ or the increase from $B$ is a very slow growing function of $\lambda$, such as $\log(\lambda)$. In such situations choosing larger values of $\lambda$ leads to improvement in the higher order terms of the regret bound, while making the lower order term (i.e. $L_{B'}\|f\|_1'\lambda R(T)$) worse; overall the regret bound can be improved by choosing a suitably large scaling factor $\lambda$ to balance between the two.

### C.3   Improvements for batch boosting

Our algorithmic technique can be used to improve convergence speed for batch boosting as well, in the setup considered by Zhang and Yu [24]. Since the focus of this paper is on online boosting, we give a high level comparison of the bounds here, making some simplifying assumptions to ease the technical details, using our notation as much as possible.

In the setup of Zhang and Yu [24], we have a base set of real valued functions $\mathcal{F}$, which we assume is symmetric and contains the zero function, $\mathbf{0}$. Then $\mathrm{span}(\mathcal{F})$ is a linear function space, and let $\|\cdot\|$ be some norm defined on $\mathrm{span}(\mathcal{F})$. For clarity of presentation, we assume that for any $f \in \mathcal{F}$, we have $\|f\| \leq 1$. This implies that for any $f \in \mathrm{span}(\mathcal{F})$, we have $\|f\| \leq \|f\|_1$.

The goal is to minimize a given convex functional $\ell : \mathrm{span}(\mathcal{F}) \to \mathbb{R}$ over its domain, $\mathrm{span}(\mathcal{F})$. We assume, for simplicity, that $\ell$ is $\beta$-smooth over $\mathrm{span}(\mathcal{F})$ under the norm $\|\cdot\|$, i.e. for any $f, f' \in \mathrm{span}(\mathcal{F})$, we have

$$\ell(f') \leq \ell(f) + \nabla\ell(f) \cdot (f' - f) + \frac{\beta}{2}\|f - f'\|^2.$$

Zhang and Yu [24] assume[4] that we have access to a base learning algorithm $\mathcal{A}$ that, given any $f \in \mathrm{span}(\mathcal{F})$ and a step size $\eta \geq 0$ can find a function $g \in \mathcal{F}$ that minimizes $\ell(f + \eta g)$. We denote the output of $\mathcal{A}$ by $\mathcal{A}(f, \eta)$.

Given such a base learning algorithm, and a sequence of step sizes $\eta_1, \eta_2, \ldots$, the boosting algorithm of Zhang and Yu [24] computes a sequence of functions $f_0, f_1, f_2, \ldots \in \mathrm{span}(\mathcal{F})$ via greedy fitting as follows: $f_0$ is set to $\mathbf{0}$, and for any $i \geq 1$,

$$f_i := f_{i-1} + \eta_i \mathcal{A}(f_{i-1}, \eta_i).$$

Define $s_0 = 1$ and $s_i = s_{i-1} + \eta_i$ for any $i \geq 1$.

For any $f \in \mathrm{span}(f)$, for $i = 1, 2, \ldots$, let $\Delta_i = \ell(f_i) - \ell(f)$ denote the optimization errors of the function $f_i$. Zhang and Yu [24] prove that for any $N \in \mathbb{N}$, we have

$$\Delta_N \leq \frac{s_0 + \|f\|_1}{s_N + \|f\|_1}\Delta_0 + \sum_{i=1}^{N} \frac{s_i + \|f\|_1}{s_N + \|f\|_1} \cdot \frac{\beta}{2}\eta_i^2. \tag{10}$$

Using the techniques in this paper, we can define a different boosting algorithm which works as follows. Given the same sequence of step sizes $\eta_1, \eta_2, \ldots$ as above, we set $f_0 = \mathbf{0}$, and for any $i \geq 1$,

$$f_i := (1 - \sigma_i\eta_i)f_{i-1} + \eta_i\mathcal{A}(f_{i-1}, \eta_i),$$

where

$$\sigma_i := \begin{cases} 1 & \text{if } \nabla\ell(f_{i-1}) \cdot f_{i-1} \geq 0 \\ 0 & \text{otherwise.} \end{cases}$$

We can analyze this algorithm along the lines of the proof of Theorem 1. First, let $g_i = \mathcal{A}(f_{i-1}, \eta_i)$. Then for $g \in \mathcal{F}$, we have $\ell(f_{i-1} + \eta_i g_i) \leq \ell(f_{i-1} + \eta_i g)$, and by the convexity and $\beta$-smoothness of $\ell$, we conclude that

$$\nabla \ell(f_{i-1}) \cdot g_i \ \leq \ \nabla \ell(f_{i-1}) \cdot g + \frac{\beta}{2}\eta_i.$$

Using this fact and following the proof of Theorem 1, we get the following bound on the optimization error $\Delta_i = \ell(f_i) - \ell(f)$ of the function $f_i$:

$$\Delta_N \ \leq \ \exp\left(-\frac{s_N - s_0}{\|f\|_1}\right)\Delta_0 + \sum_{i=1}^{N} \exp\left(-\frac{s_N - s_i}{\|f\|_1}\right) \cdot \frac{\beta}{2}\eta_i^2(s_i^2 + 1). \tag{11}$$

We can compare our bound (11) to the bound (10) of Zhang and Yu [24], by comparing corresponding terms in the bound. For each term, we can calculate how large $s_N$ needs to be for the term to be reduced to less than some given bound $\epsilon$. To reduce the first term to less than $\epsilon$ our algorithm needs $s_N \geq \|f\|_1 \log(\frac{\Delta_0}{\epsilon}) + s_0$, whereas the algorithm of Zhang and Yu [24] needs $s_N \geq (\frac{\Delta_0}{\epsilon})(s_0 + \|f\|_1) - \|f\|_1$. As for the second term, to reduce the $i$-th term in the sum to less than $\epsilon$, our algorithm needs $s_N \geq \|f\|_1 \log(\frac{\beta\eta_i^2(s_i^2+1)}{2\epsilon}) + s_i$, whereas the algorithm of Zhang and Yu [24] needs $s_N \geq (\frac{\beta\eta_i^2}{2\epsilon})(s_i + \|f\|_1) - \|f\|_1$. Since in either case, the dependence on $\epsilon$ is $\log(\frac{1}{\epsilon})$ for our algorithm, whereas it is $\frac{1}{\epsilon}$ for the algorithm of Zhang and Yu [24], we conclude that our algorithm converges exponentially faster.

# D   Description of Data Sets and Detailed Experimental Results

The datasets come from the UCI repository and various KDD Cup challenges. Below, $d$ is the number of unique features in the dataset, and $s$ is the average number of features per example.

| Dataset | Number of instances | Total number of features | Average number of features per example | Task | Label range |
|---|---|---|---|---|---|
| a9a | 48,841 | 123 | 14 | classification | $[-1, 1]$ |
| abalone | 4,177 | 10 | 9 | regression | $[1, 29]$ |
| activity | 165,632 | 20 | 18.5 | classification | $[-1, 1]$ |
| adult | 48,842 | 105 | 12 | classification | $[0, 1]$ |
| bank | 45,211 | 45 | 15 | classification | $[-1, 1]$ |
| cal_housing | 20,640 | 9 | 9 | regression | $[0, 1]$ |
| casp | 45,730 | 10 | 10 | regression | $[0, 1]$ |
| census | 299,284 | 401 | 32 | classification | $[-1, 1]$ |
| covtype | 581,011 | 54 | 12 | classification | $[-1, 1]$ |
| kddcup04 (phy) | 50,000 | 74 | 32 | classification | $[0, 1]$ |
| letter | 20,000 | 16 | 15.6 | classification | $[-1, 1]$ |
| shuttle | 43,500 | 9 | 8 | classification | $[-1, 1]$ |
| slice | 53,500 | 385 | 135 | regression | $[0, 1]$ |
| year | 463,715 | 90 | 90 | regression | $[0, 1]$ |

The following table provides the online squared losses summarized in Section 5.

| Dataset | SGD | | | Regression stumps | | | Neural Networks | | |
|---|---|---|---|---|---|---|---|---|---|
| | Baseline | Alg 1 | Alg 2 | Baseline | Alg 1 | Alg 2 | Baseline | Alg 1 | Alg 2 |
| kddcup04/phy | 0.7475 | 0.7466 | 0.7470 | 0.9201 | 0.7733 | 0.7924 | 0.7441 | 0.7480 | 0.7446 |
| cal_housing | 0.0094 | 0.0094 | 0.0104 | 0.0151 | 0.0138 | 0.0124 | 0.0096 | 0.0096 | 0.0107 |
| casp | 0.0632 | 0.0631 | 0.0630 | 0.0741 | 0.0741 | 0.0742 | 0.0639 | 0.0632 | 0.0631 |
| a9a | 0.4261 | 0.4283 | 0.4249 | 0.5749 | 0.5074 | 0.5758 | 0.4256 | 0.4266 | 0.4246 |
| abalone | 3.7263 | 3.7482 | 3.7154 | 6.7791 | 3.8273 | 4.2270 | 3.7380 | 3.7255 | 3.7212 |
| activity | 0.0334 | 0.0337 | 0.0316 | 0.4492 | 0.1454 | 0.3141 | 0.0192 | 0.0143 | 0.0186 |
| adult | 0.1055 | 0.1057 | 0.1056 | 0.1388 | 0.1261 | 0.1250 | 0.1081 | 0.1062 | 0.1081 |
| bank | 0.2971 | 0.2968 | 0.2973 | 0.3774 | 0.3240 | 0.3257 | 0.2962 | 0.2969 | 0.2969 |
| census | 0.1544 | 0.1545 | 0.1553 | 0.2073 | 0.1884 | 0.1789 | 0.1531 | 0.1531 | 0.1523 |
| covtype | 0.7256 | 0.7270 | 0.7286 | 0.7910 | 0.7986 | 0.7911 | 0.6807 | 0.6465 | 0.6757 |
| letter | 0.6441 | 0.5698 | 0.6108 | 0.7420 | 0.7087 | 0.7168 | 0.6542 | 0.5729 | 0.6108 |
| shuttle | 0.1616 | 0.1547 | 0.1577 | 0.8551 | 0.3678 | 0.4354 | 0.0760 | 0.0694 | 0.0802 |
| slice | 0.0076 | 0.0067 | 0.0065 | 0.0559 | 0.0362 | 0.0410 | 0.0054 | 0.0022 | 0.0044 |
| year | 0.0116 | 0.0119 | 0.0115 | 0.0152 | 0.0140 | 0.0141 | 0.0116 | 0.0119 | 0.0122 |

## Footnotes

[4]This is a slight simplification of the base learning algorithm considered in [24], which also performs a search over the step size $\eta$. Also, the analysis in [24] allows some optimization error for the base learning algorithm; to simplify the comparison we assume this error is 0.