[Reviews · NeurIPS 2015]

Submitted by Assigned_Reviewer_1

The paper explores online gradient boosting regression, providing theoretical guarantees on its performance.

I am not an expert in theoretical analysis of learning methods, and therefore I focus here on the experimental evaluation of section 6.

I find the experiments interesting, showing that the proposed approach can improve over the online weak learners used. However, to compare against the analogous batch method, it would be interesting to add the results of batch boosting on the same data, to assess their relative performance.
Summary: I am not an expert in the theoretical analysis of learning methods, and therefore I focus here on the experimental evaluation of section 6. It would be also interesting to compare the online learning methods against a batch version of boosted regression.

Submitted by Assigned_Reviewer_2

This paper proposes an online gradient boosting algorithm for regression. The corresponding regret bound is derived theoretically under the settings of considering (i) the span of the D-bounded reference class of functions and (ii) the convex hull of the function class. The algorithm is evaluated experimentally using a set of data sets as detailed in the supplementary material. It seems that the boosting only got more significant improvement on some data sets e.g., covtype, letter, shuttle, slice with SGD and NN being the base learner. In the literature, there also exist other works on online boosting for regression (e.g., Babenko, B.; Ming-Hsuan Yang; Belongie, S., "A family of online boosting algorithms," Computer Vision Workshops (ICCV Workshops), 2009 IEEE 12th International Conference on , vol., no., pp.1346,1353, Sept. 27 2009-Oct. 4 2009) and it is not sure how this work is compared them.
Summary: This paper proposes an online gradient boosting algorithm for regression. The corresponding regret bound is derived theoretically and the algorithm is evaluated experimentally with some performance improvement in terms of accuracy.

Submitted by Assigned_Reviewer_3

The authors study the online setting for boosting in the context of regression problems.

Specifically, they describe and analyze two algorithms for online boosting for regression: (1) a boosting algorithm that uses a linear span of the base learning functions as the prediction function (i.e., the standard case) and (2) a boosting algorithm that uses a convex hull (CH) of the base functions as the prediction function.

Algorithm (1) more closely aligns with existing gradient boosting approaches and provides the most practical insight.

Algorithm (2) has some nice theoretical properties with respect to being optimal for the specified setting (and may give some insight to optimality in the span(F) case of algorithm (1).

Experiments are also performed on 14 standard datasets and show that the proposed approaches outperform the base learners on average (and nearly universally when looking at the supplementary material).

While I would like to see more empirical results (that I think are fairly achievable), this paper combined with [Beygelzimer, et al., ICML15] would represent the state of the art in online boosting methods, both theoretically and practically.

Quality: [well motivated, good math in terms of being hard problems, would like a comparison to existing work]

From a theoretical perspective, I like this paper quite a bit.

They consider a difficult setting that has significant practical implications and push forward the theory in a non-trivial way by computing the regret bound for both the span(F) and CH(F) settings, demonstrating that Algorithm (2) is optimal (up to constant factors) for the CH setting.

Furthermore, Section 5 discusses some practical concerns and demonstrates that the proposed online boosting algorithm has superior convergence properties to the batch setting (which could be important in large scale settings).

The weakest aspect of this paper is the experiments.

Firstly, while I understand it is difficult to fit some of the results from the supplemental material in the body of the paper, the results presented in the paper are just not that insightful (while I like those in the supplementary material - it would be great if some of these could make their way into the paper).

However, it would also be a stronger paper if there was a comparison to existing regression boosting techniques, especially since it isn't difficult to perform such experiments (including possible convergence results as this seems to be a point).

Clarity:

In all honesty, this is a somewhat difficult paper that required a fair amount of background material before I really understood the claims and significance of the results.

The supplementary material helped quite a bit.

However, I don't see how this could be overcome.

I think that Sections 1.1 and 5 can be tightened up a bit to expand section 6.

However, this is a case where the conference paper will have to be a summary of a longer paper - which is completely fine given the depth of the results presented.

Originality:

As stated by the authors (and verified to the best of my ability), this is the first thorough analysis of online boosting for regression problems.

This is a non-trivial step in terms of the resulting algorithm, analysis techniques, and state-of-the-art for boosting.

Significance:

While boosting is primary used for classification (especially in online settings), this paper is ostensibly a step toward putting increasing the use of boosting algorithms for regression on large data sets (where random forests seems to be the dominant ensemble approach).

Furthermore, the convergence results as compared to the batch boosting for regression setting is useful in the general case and will likely provide insight with respect to improving the batch setting.

Furthermore, while there has been some recent work on Frank-Wolf, convex hulls, etc. with respect to boosting algorithms, I think the authors have put together some of these ideas in an approachable way that at least affected the way I think about these problems.

My one criticism is that it doesn't seem that Algorithm (2) is all that useful outside of the theoretical discussion (which is still worthwhile).

Also, without comparisons to the batch setting, it isn't entirely clear how effective the proposed methods are in practice.
Summary: The authors explore online boosting for regression -providing a usable algorithm, regret bounds, and good preliminary empirical results.

By understanding this paper and [Beygelzimer, et al., ICML15], I think researchers would have a pretty good picture of the state-of-the-art in online boosting.

Therefore, I think it merits being accepted.

Submitted by Assigned_Reviewer_4

I would suggest removing Section 5 and include more experiments instead.
Summary: Theoretically a strong contribution but more experiments would have been welcome.

Submitted by Assigned_Reviewer_5

The paper extends the theory of boosting to the online learning setting, proposing strong learning algorithms, building up from weak learners.

First of all, I want to say that the topic of online boosting is quite fascinating. Theory of boosting has been of great significance to both statistics and machine learning community over the past two decades. Recently, there has been a couple of papers which has tried to extend boosting for binary classification to the online setting. In fact, the latest paper (by Satyen Kale et al.) won the best paper award at ICML 2015!

Significance- As discussed above, the significance of online boosting is beyond question.

Originality- As far as I know, no one has published work on extending boosting for regression to the online setting. Definitely original.

Quality- Assuming the proofs are correct (I have not checked the analysis of Algo.2 and I have a major concern about the regret bound of Alg.1, which I will come to later), the paper is definitely of good quality.

Clarity- In my opinion, the main drawback of the paper is the presentation. I will list as many concerns as I can recollect. 1. Discussion about weak and strong learners- The paper directly gives a definition of weak learning algorithm and strong learning algorithm. What are the intuition behind them? How do they relate to weak and strong learning for regression in the offline case? I am much more familiar with boosting for classification. I know that boosting for regression is approached more from a statistical viewpoint (as greedy forward model selection). Is there a weak and strong learning equivalent in the offline setting for regression?

2. Line 117-125- I am slightly perturbed that just 1 paragraph sums up online boosting for regression. It is a very novel concept and in my opinion, requires much more discussion. Why is the strong learner defined for richer regression function class F' and richer loss function class C'? Specifically, why is the loss class changing? What is the equivalent in offline setting?

3. Line 136-138- I did not understand why we cannot have uniform regret bound for all functions in span(F). In fact, if the paper is the online equivalent of Zhang and Yu's paper (as stated by the authors), then the strong learner should converge to the best linear combination of base functions (i.e., min over span(F)).

4. The richer loss function class C not only needs to be convex but also Lipschitz and more importantly, smooth. This was not mentioned in the abstract, where it seems the results will hold for all convex functions. 5. Alg.1- Why not have some intuition leading to Alg.1? I had to read the proof to even start understanding what is going on in the algorithm. It is not necessarily a criticism, but I always feel discussions after a very technical algorithm is helpful. As an example, see Kale's paper on online boosting. They lead to their algorithm in a very natural way. 6. Theorem 1- I have a big concern about the theorem. First of all, it seems for the regret to go to 0, N -> \infty. However, for very large N, does the algorithm remain efficient? This is because in step 8, d-dimensional vectors y^i are created for all copies of A (i.e. total of N). Will not soon become too large?

More important question- The claim is that average regret goes to 0 as N and T-> \infty. It is not clear what is the rate at which the regret goes to zero. Considering the first term in the regret will go to zero only for \eta= O(log(N)/N) (though it is not clear to me at what rate), the second term is then going to 0 at rate O(log(N)/N), which is slower than even linear (in N) rate. A major advantage of boosting is the rate at which training error goes to zero. Even for online boosting (Kale et al.'s paper), they have error going to zero exponentially fast with N (effectively agreeing with offline boosting, where training error goes to zero exponentially fast with N). So combined with the fact that the regret is not even uniform over the linear span, I would like some clarification of the issue.

7. In Alg.1, why cannot we just set B=\etaND? Why need to take a min over some other value? I did not see what it was adding to the algorithm. Line 284 shows the equality and the upper bound in line 297 will come from the Taylor expansion. Though taking a smaller B will give smaller regret, on the other hand, B= \etaND will get rid of the extra \eta^2 \beta B^2 term and make it more readable, isn't it? 8. Why will we care about competing with the convex hull of H? What is the offline equivalent? I can understand competing with best in span(F) but why convex hull?

8. In Theorem 2., what does dependence of this regret bound on N is optimal mean? Should I interpret the sentence as : No strong learning algorithm can have a tighter regret in terms of N, against all adversary?

There are more places where I felt more discussion could have been added. While I want to emphasize that I strongly believe this is a good piece of work, I think it will be greatly beneficial to the machine learning community if the paper goes through a solid round of editing before acceptance.

Summary: I believe that the paper deals with a very interesting topic and will be of genuine interest to the community. However, the paper needs thorough polishing and lot of modification to the presentation and discussion of results before it can be published.

Author Feedback
Author rebuttal: We thank all reviewers for their careful reading of the paper. We give a few general points of clarification below, followed by individual responses to some reviews.

1. Presentation: We will shorten/remove some material to include more discussion and introduction of the background material, making the paper more self-contained and easier to read.

2. The significance of Alg 2:
(a) Alg 2 may still be useful in practice when a bound on the norm of the comparator function is known in advance, using the observations in section 5.2,
(b) the analysis of Alg 2 is cleaner and easier to understand (than that of Alg 1) for readers who are familiar with the Frank-Wolfe method, and this serves as a foundation for the analysis of Alg 1
(c) the lower bound we proved for competing with the convex hull elucidates the achievable convergence rates in terms of N, the number of weak learners: it shows that at best the convergence rate is 1/N.

3. Convergence rates in terms of N: Even in the offline case, (e.g. Zhang, Yu 2005 [ZY] or Mason et al, 1999 [MBBF]) convergence to the best function in the span only happens as the number of boosting steps (which corresponds to N in our setting) goes to infinity. As for rates, exponential convergence rate is possible in the classification setting (both batch and online) because the assumption there is much stronger: viz. the weak learner has a noticeable margin gamma > 0 over random guessing. Similar exponential convergence rates can be obtained in the regression setting as well making a similar strong assumption on the weak learner. However, this assumption becomes too unnatural in the regression setting, and we chose to present our results essentially making no assumptions on the weak learner (beyond online ERM) exactly as in [ZY]. Furthermore, our lower bound shows that the best convergence rate we can hope for is 1/N. Our convergence rates are actually better (see section 5.3) than the ones proved in [ZY] for the offline setting. To summarize, the convergence rates we obtain are better than the best previously known rates for boosting in the regression setting.

4. Experiments: While we agree that a comprehensive experimental comparison with other methods (in particular, batch gradient boosting) would greatly complement this paper, it was not its focus. The main question that we wanted to address with our experiments was whether it is possible to boost effectively in an online fashion in practice. Our proof-of-concept experiments show that the answer is positive, using a theoretically justified algorithm. We emphasize that our algorithms are truly online, and thus apply in many settings where batch algorithms simply don't, e.g., when new data keeps coming in or the best predictor changes over time.

Response to Assigned_Reviewer_4:
1. Definitions of weak and strong learners: We will make it clear that our definitions of weak and strong learner are exactly the online analogues of the definitions in the offline setting. In the offline setting, a weak learner is understood to be an ERM oracle (as in [ZY]) or, in the case of gradient boosting (as in [MBBF]) as an ERM oracle with linear losses, over the base function class. Our definition simply changes the ERM oracle in the offline setting to an online learning/ERM oracle.

2. Richer classes: The reason we have a larger function class F' and loss function class C' is to explicitly acknowledge the generality of the approach; offline boosting algorithms also have these properties without being explicitly mentioned. We agree that we should have mentioned that the loss functions need to be smooth in the abstract; we will fix this in the final version. However it is easy to smoothen non-smooth functions using perturbations and thereby obtain results for non-smooth functions as well. Furthermore, if the weak learner is deterministic, in Section 2 we mention that our algorithms directly work for non-smooth loss functions.

3. Uniform regret bound for span(F): In the offline case as well, the convergence rate (see [ZY]) to any given function in the span depends on its norm. Uniform bounds are not possible since we are competing with unbounded functions.

4. Setting the parameter B: we can certainly always set B = \eta ND, but we chose to have the more complicated setting because for loss functions such as p-norm loss, modified least squares, etc, it allows us to obtain a smaller value of B (see section 4), improving the regret bound.

5. Intepretation of lower bound in Theorem 2: This is exactly what you said.

Response to Assigned_Reviewer_5:
Relation to other literature: To our knowledge, ours is the first paper to have a theoretical analysis of online boosting for regression. The paper by Babenko et al is a heuristic with no theoretical guarantees. Due to lack of space we haven't surveyed such heuristic online boosting algorithms.